

# Epidemiology of tuberculosis and treatment outcomes among children in Pakistan: a 5 year retrospective study

Madeeha Laghari[1], Syed Azhar Syed Sulaiman[1], Amer Hayat Khan[1] and Naheed Memon[2]

[1] Department of Clinical Pharmacy, Universiti Sains Malaysia, Minden, Penang, Malaysia
[2] College of Pharmacy, Liaquat University of Medical & Health Sciences, Jamshoro, Pakistan

## ABSTRACT

**Background**. Regardless of the advancement in medical technologies, the diagnosis of tuberculosis (TB) in children has remained a challenge. Childhood TB is rampant and an important cause of morbidity and mortality. The objective of this study was to determine the trend of TB and treatment outcomes in children aged ≤14 years registered for TB treatment under DOTS course in three districts of Sindh, Pakistan.

**Methods**. For this retrospective study, records of TB children (≤14 years) registered for the treatment of TB from January 2011 to December 2015 in three districts of Pakistan, were collected. Demographic data, baseline weight, clinical manifestations, radiography, histopathology results and treatment outcomes were collected from TB unit registers.

**Results**. A total of 2,167 children were treated for TB during the study period. Of these, 1,199 (55.3%) were females and 1,242 (57.3%) were from urban areas. Over three-quarter of patients (76.9%) had pulmonary TB with 13.3% of sputum smear positive cases. The overall treatment success rate was 92.4%. In multivariate analysis, rural residents (OR: 2.146, $p < 0.001$), sputum smear positive cases (OR: 3.409, $p < 0.001$) and re-treated patients (OR: 5.919, $p < 0.001$), were significantly associated with unsuccessful treatment outcomes. However, age group ≤2 years, male and those who were underweight were found to have the highest risk of pulmonary tuberculosis (OR: 1.953, $p < 0.001$; OR: 1.262, $p = 0.028$; OR: 1.342, $p = 0.008$), respectively.

**Conclusion**. Patients at risk of treatment failure must be given particular attention. Moreover, strategies are needed to further improve the diagnosis and treatment of TB among children and improve the recording system.

# BACKGROUND

Tuberculosis (TB) is the ninth leading cause of death at a global level, ranking higher than Human Immunodeficiency Virus/Acquired Immune Deficiency Syndrome (HIV/AIDS). During 2016, 10.4 million new TB cases were detected worldwide, of which 90% adults and 65% were male. During the same year, children aged <15 years comprised up to 6.9% of the notified new TB cases (*WHO, 2017*). According to a WHO report, a continuous rise has been seen in case detection rate of childhood TB worldwide with a parallel increase in

Corresponding author
Madeeha Laghari,
madeehalaghari@gmail.com

the death toll of children with TB. In mathematical modelling study, it was estimated that 239,000 children died from TB in 2015 (*Dodd et al., 2017*).

Childhood TB has traditionally been unnoticed by practitioners, researchers, and experts partially, because of the notion that children are rarely infectious and therefore add little to the dissemination of disease (*Adams et al., 2014*). Without successful detection and treatment of TB infection both in adults and children, eradication strategies will be unproductive and attempts at epidemic TB control will be failed in future. This is because children present the repository for new cases to develop in future (*Seddon & Shingadia, 2014*).

However, concern about childhood TB has increased significantly and in 2012, an annual WHO report included an estimation of childhood TB was included for the first time (*WorldHealthOragnization, 2012*). Childhood TB is rampant and an important cause of morbidity and mortality in developing countries owing to poor socio-economic conditions, starvation, overcapacity, HIV co-infection (*Glaziou et al., 2013*) and the high prevalence of TB in adults contacts (*Tilahun & Gebre-Selassie, 2016*).

Pakistan stands among the eleven high TB burden countries and was one of the six countries that stands out as having the largest number of cases in 2014 (*World Health Oragnization, 2015*). In 2016, of the total estimated incidence, 518,000 TB cases were notified in Pakistan, of which 51,000 were cases of children aged ≤14 years. Pakistan together with India, Indonesia, China, and Philippines accounted for 56% of the global total incident cases in 2016 (*WHO, 2017*).

Sindh is a multicultural province of Pakistan and is portrayed by an extensive gap between rich and destitute individuals with unequal access to health services. Inhabitants of low-wage neighbourhoods, for instance, suffer from overcrowding and malnutrition. Consequently, they are susceptible to developing TB (*Akhtar, Carpenter & Rathi, 2007*). In Pakistan, research contribution in childhood TB is still narrow and only few studies have been conducted until now. However, to the best of our knowledge this was the first study to discuss proportion of childhood TB cases treatment outcomes among children in Sindh province, particularly in studied districts.

The objectives of the study were to describe the trend of TB and treatment outcomes with the risks of treatment failure in children aged ≤14 years.

## METHODS

### Ethics approval and consent to participate

Ethical approvals were issued by the relevant Institutional Research and Ethics Boards (IREBs) of Shah Bhitae Hospital Latifabad, Hyderabad Liaquat University Hospital Hyderabad / Jamshoro, Sindh Government Hospital Qasimabad, Hyderabad, Sayed Baqadar Shah Civil Hospital Matiari and Institute of chest diseases Kotri Sindh, Pakistan, (Vide Letter No: SBGH/L.ABAD HYD-1575; Dated: 13-04-17, LUH/Estt/-23176/14; Dated: 06-08-2016, MS-SGHQ/HYD/2187: Dated:13-04-17, CS/CH/MAT:1761; Dated:18-05-16 and ICDK/771; Dated: 12-04-17, respectively).

## Study design and data collection

This was a retrospective study. Data was collected from January 2011 to December 2015 for children suffering from TB and registered under DOTS. In order to determine the proportion of childhood cases, total TB cases including the numbers of adults were collected from TB registers. The registers contain information regarding the socio-demography (age, gender, and residence), clinical manifestations and laboratory examination (sputum or gastric aspirates microbiology, HIV status, radiography and histopathology results). All that information was transferred on a data collection form specially designed for the study. Patients were grouped as ≤2 years, 3 to 5 years, 6 to 10 years and 11 to 14 years. Body weight was calculated in percentiles using the data table of weight for age charts by Centres for Disease Control and Prevention (CDC). Patients with weight <5 percentiles were recorded as underweight and those with ≥95 percentiles were considered as overweight (*Centers for Disease Control and Prevention NCfHS, 2001*).

Children with TB are diagnosed and treated according to NTP under the DOTS strategy (Fig. 1). Each child with clinical symptoms including a cough lasting for ≥2 weeks, fever, night sweats, dyspnoea and sputum production were examined. Assessment of nutritional status was done based on weight-for-age data from weight-for-age charts (*Centers for Disease Control and Prevention, 2001*). Children were diagnosed as having TB by considering suggestive clinical features, the history of contact, positive TST (≥10 mm was considered positive) (*WHO, 2014*), scoring charts (suggested by the Pakistan Paediatric Association) and evidence of TB in chest X-ray (CXR) for pulmonary TB (PTB). Culture and Xpert MTB/RIF assays were used as add-on tests and specifically performed to exclude resistant TB. Furthermore, common forms of EPTB were diagnosed based on a positive result of an appropriate test: peripheral lymphadenitis: lymph node biopsy or fine needle aspiration (FNAC); miliary TB: CXR; TB meningitis: lumbar puncture with cerebrospinal fluid (CSF) analysis or cerebral computed Tomography (CT); pleural effusion TB: CXR and pleural tap; abdominal TB: abdominal ultrasound and ascetic tap; and bones/joints TB: X-ray, joint tap or synovial biopsy (*National TB Control Program, 2015*).

The regimen was prescribed on a category basis. The four treatment categories are as follows: Category-I is new smear positive PTB (PTB+), severe forms of new extra-pulmonary TB (EPTB), new severe concomitant HIV disease and TB meningitis. Category-II is previously treated PTB+ (relapse, treatment after interruption and treatment failure). Category-III comprises new smear negative PTB (PTB-) and less severe forms of EPTB. Category-IV includes chronic and multidrug resistant TB (MDR-TB). The children diagnosed with TB always have an intensive phase of 2 to 3 months and a continuation phase of 4 to 6 months. The anti-TB drugs dosage depends on body weight and category of patient.

Treatment of children with TB aged ≤14 years as per NTP is given in Table 1. NTP follows the treatment and dosage criteria as recommended by WHO (*WHO, 2010*). Patients weighing <5 kg are treated with individualized dosages while those weighing more than 30 kg are treated using adult dosages.

According to the WHO (*World Health Oragnization, 2014*), patients who were stated "cured" and/or had "completed treatment" were termed as a "treatment success", and

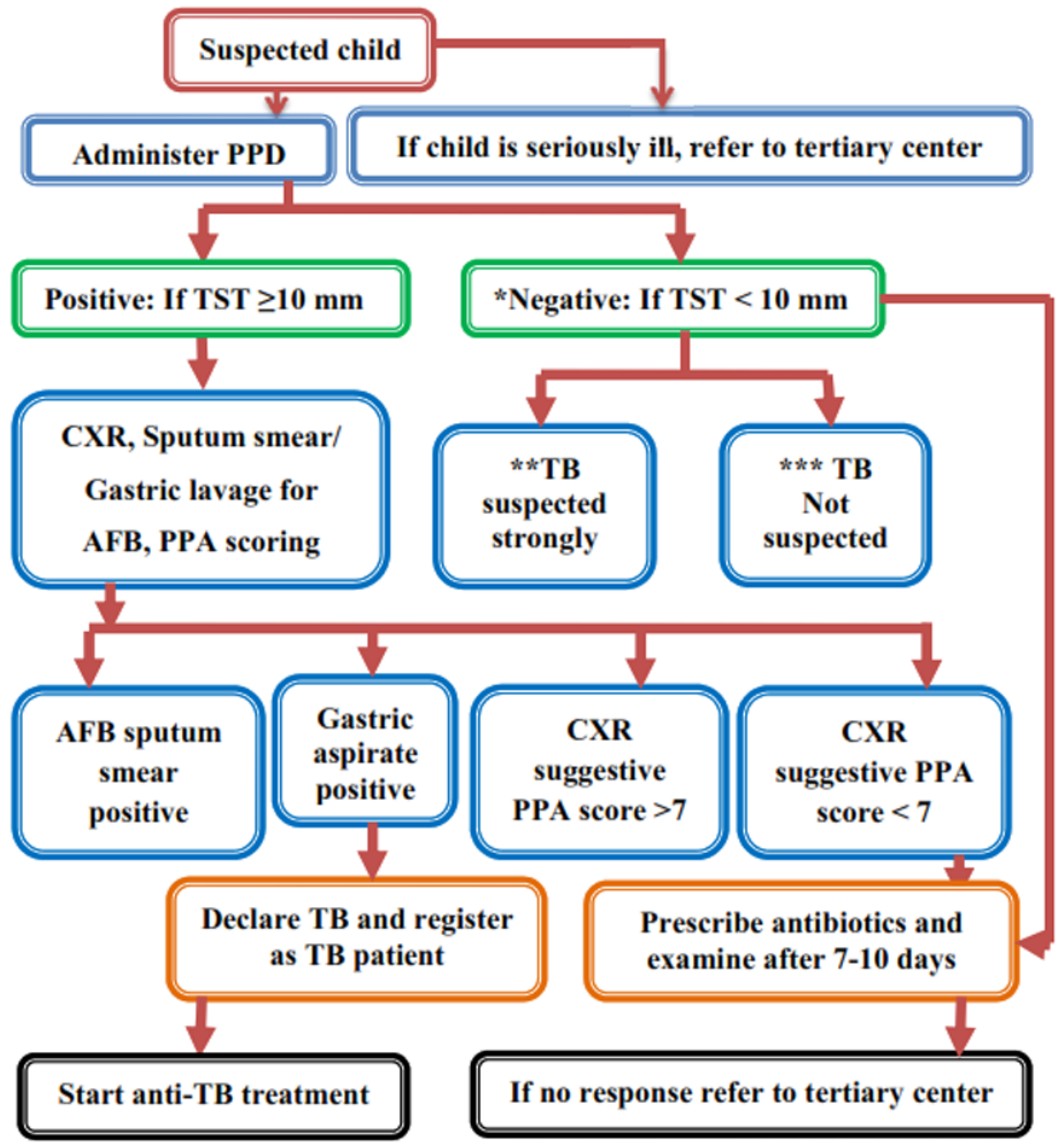

**Figure 1   NTP flow chart for assessment of a child with suspected TB.** *, Negative TST does not rule out TB, evaluate further if strongly suspected. **, prolong illness like cough, fever, weight loss, abdominal mass, lymphadenopathy and history of contact with TB case in family. ***, Absence of above.

all those patients who had defaulted, died, or experienced treatment failure were reported under the category of "unsuccessful treatment". A *cured* patient was a bacteriologically confirmed PTB patient and became smear negative in the last month of treatment and on at least one previous occasion. *Treatment completed* included TB patients who completed treatment with no record to show smear negative in the last month of treatment and on at least one previous occasion.

**Table 1  Dosage of first-line anti-TB drugs for children based on NTP and WHO.**

| Recommended daily dose for first line anti-TB drugs for children | | |
|---|---|---|
| **Anti-TB drugs** | **Dose and range (mg/kg)** | **Maximum dose (mg)** |
| Isoniazid (H) | 10 (7–15) | 300 |
| Rifampicin (R) | 15 (10–20) | 600 |
| Pyrazinamide (Z) | 35 (30–40) | – |
| Ethambutol (E) | 20 (15–25) | – |

| Recommended treatment regimens for TB in children | | |
|---|---|---|
| **TB diagnostic category** | **Anti-TB drug regimens** | |
| | **Intensive phase** | **Continuous phase** |
| Smear negative pulmonary TB | | |
| Intrathoracic lymph node TB | 2RHZ | 4RH |
| Tuberculosis peripheral lympadenitis | | |
| Extensive pulmonary TB | | |
| Smear positive pulmonary TB | | |
| Severe forms of extra-pulmonary TB | 2RHZE | 4RH |
| (other than TB meningitis/osteoarticular TB) | | |
| TB meningitis/osteoarticular TB | 2RHZE | 10RH |

*Defaulters patients* were those whose treatment was interrupted for 2 consecutive months or more for any reason and *failure patients* were those whose sputum smear was positive at 5 months of treatment or later.

## Data analysis

Data was entered and analysed using the Statistical Package for the Social Sciences (SPSS) for Windows, version 24. Frequencies of categorical variables were generated by descriptive statistical methods. Comparisons in categorical groups were performed using the chi-square test. A logistic regression model was used to determine the predictors of unsuccessful outcome and PTB. $P$ values $<0.05$ were considered statistically significant.

# RESULTS

## Proportion and trend of notification cases of childhood TB

Figure 2 shows the proportion of childhood TB cases at the study site. During 2011 to 2015, childhood TB accounted 11.3% (216719219) of all TB burden in three districts. The Proportion of childhood TB ranged from 8.5% in 2011 to 18% in 2015. In 2012, 2013 and 2014, the child cases represented 8.5%, 8.8% and 9.9% respectively, of overall TB cases at the study site. Differences in the proportion of childhood TB between the districts were observed, with the highest being in Jamshoro district and the lowest in the Matiari district.

A total of 2,167 patients meeting the inclusion criteria were enrolled in the study. In Hyderabad, childhood TB cases are registered at Sindh Government Hospital Qasimabad, Hyderabad (SGH-QH), Shah Bhitai Government Hospital, Latifabad, Hyderabad (SBGH-LH) and Liaquat University of Medical and Health Sciences Civil Hospital, Hyderabad
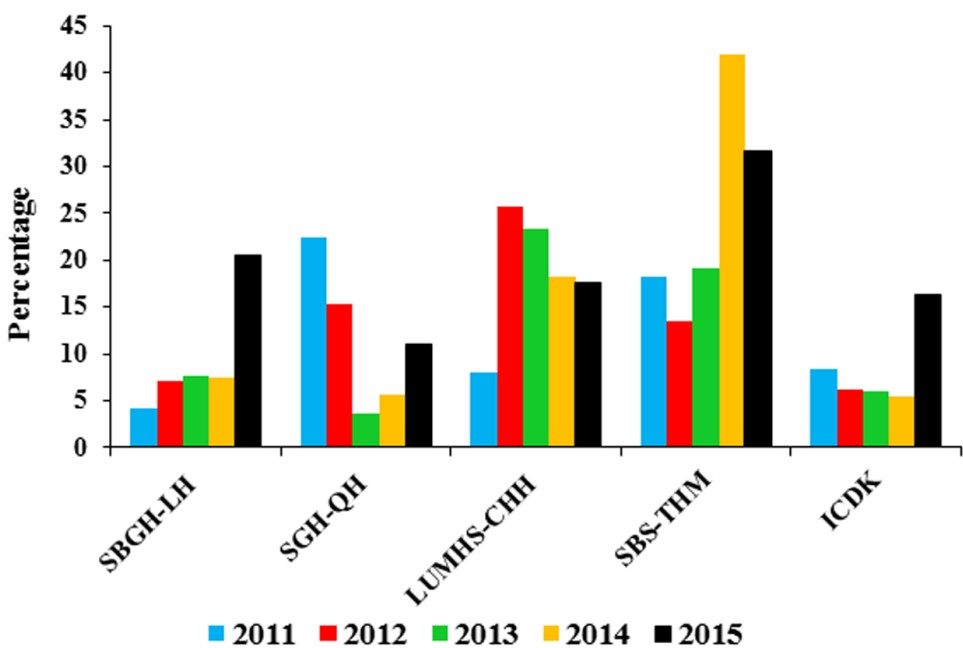

**Figure 2** **Proportion of childhood TB burden in five hospitals (2011–2015).** SBGH-LH, Shah Bhitai Government Hospital, Latifabad, Hyderabad; SGH-QH, Sindh Government Hospital Qasimabad, Hyderabad; LUMHS-CHH, Liaquat University of Medical and Health Sciences Civil Hospital Hyderabad; SBS-THM, Syed Baqadar Shah Taluka Hospital, Matiari; ICDK, Institute of Chest Diseases Kotri, Sindh. The figure shows of childhood TB cases in all five hospitals. The trend varied greatly for all these hospitals during study time. SBGH-LH and ICDK had maximum cases in 2015. For SGH-QH highest cases were noted during 2011, LUMHS-CHH in 2012 and SBS-THM in 2014.

(LUMHS-CHH) where treatment is provided under the DOTS program. Among these three hospitals, the maximum numbers of patients were registered in LUMHS-CHH (487) as compared to SBGH-LH (136) and SGH-QH (52). Overall, the highest numbers of children with TB (1,172) were registered in the Institute of Chest Diseases Kotri, Sindh (ICDK) during the present study which was almost four times the number of cases (320) registered at Syed Baqadar Shah Taluka Hospital, Matiari (SBS-THM). The increased trend was observed from 2011 to 2015 in SBGH-LH and ICDK. Conversely, in SGH-QH and LUMHS-CHH, the highest numbers of cases were reported at the start of the study and then a steady decline was observed. For SBS-THM, the increasing trend was seen from 2011 to 2014, and that dropped off in 2015.

### Trend of type of TB from 2011–2015

Initially, the proportion of PTB- was 15.2% that declined to 14.7% in 2012. A slight increase was noticed in the rate of PTB- (15.4%) during 2013 that dropped to 14% in 2014 but then went up to 15.6% at the end of the period. Conversely, the proportion of PTB+ was low (12.4%) during 2011 but then rose steadily to 13.4% and 14.2% in 2012 and 2013, respectively. The rate of PTB + subsequently descended from 12.7% in 2014 to 8.7% in 2015. In terms of EPTB, the rate gradually climbed between 2011 and 2014 (21.2–29.2%) (Fig. 3) and subsequently dropped to 18.7% at the end of study period.
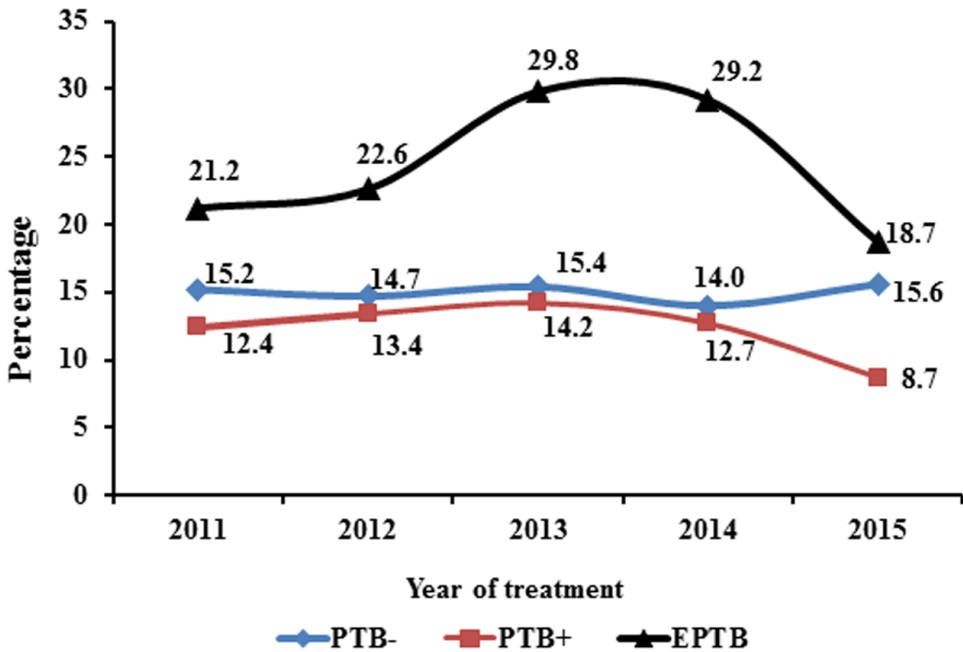

**Figure 3 Trend of type of TB during 5 years of study.** PTB−, smear negative pulmonary TB; PTB+, smear positive pulmonary TB; EPTB, extra-pulmonary TB. Trend of different types of TB were noticed for children from 2011 to 2015. There was a slight variability was seen for PTB−. The maximum fluctuation was observed for EPTB starting from 21.2% reaching up to 29.8% and then finally declining to 18.7%. A decreasing trend was seen for PTB+ among children.

## Demographic and clinical characteristics of patients

Of the total 2,167 patients, 1,199 (55.3%) were females and 968 (44.7%) were males with a mean age of 7.4 ± 4.4 years. The majority of the children were aged between 11 to 14 years (33%) followed by 28% in the age group of 6 to 10 years. Children aged 3 to 5 years were the smallest group comprising 17% of the total (Table 2). Baseline weight was documented for all patients and 75.2% of them were found to be underweight. Over three-quarter, 1,666 (76.9%) of children were diagnosed with PTB. Of the total PTB cases, 331 (15.3%) had PTB- and 221 (10.2%) presented with PTB +. The remaining 1,114 (51.4%) of PTB cases were those who had no sputum examination (PTBNS). The majority (91%) of PTB+ were ≥9 years with the remaining 9% aged 7 to 8 years. In 501 cases of EPTB, peripheral lymphadenitis was the most frequent (62.4%) followed by 17.3% with abdominal TB (Table 2).

Concerning therapeutic categorization, 2,083 (96.1%) were new cases (patients had either received no ATT or received for <1 month in the past) and 84 (3.9%) were cases of re-treatment (those received ATT >1 month in the past). Of re-treatment cases, 9 (10.7%) were registered as relapse cases, 28 (33.3%) were default, 14 (16.7%) were failure and 33 (39.3%) cases were recorded as others. Similarly, 41 (48.8%) presented with PTB+ and 20 (23.8%) had PTB- while 23 (27.4%) were cases of EPTB.

**Table 2** Socio-demographic and baseline clinical characteristics of children with TB in studied hospitals ($n = 2,167$).

| Variables | Frequency (%) |
|---|---|
| **Gender** | |
| Male | 968 (44.7) |
| Female | 1,199 (55.3) |
| **Age (years)** | |
| ≤2 | 461 (21.3) |
| 3–5 | 384 (17.7) |
| 6–10 | 607 (28) |
| 11–14 | 715 (33) |
| **Residence** | |
| Rural | 925 (42.7) |
| Urban | 1,242 (57.3) |
| **Type of TB** | |
| Smear positive PTB (PTB+) | 221 (10.2) |
| Smear negative PTB (PTB−) | 331 (15.3) |
| PTB with unknown smear (PTBNS) | 1,114 (51.4) |
| EPTB | 501 (23.1) |
| **EPTB site** | |
| Peripheral Lymph nodes | 313 (62.5) |
| Abdominal | 87 (17.4) |
| Pleural | 32 (6.4) |
| Miliary | 21 (4.2) |
| Bones/Joints | 19 (3.8) |
| Meningitis | 4 (0.8) |
| Skin | 3 (0.6) |
| Others | 22 (4.4) |
| **Weight (percentiles)** | |
| Underweight* | 1,629 (75.2) |
| Normal weight | 538 (24.8) |
| **Registration category** | |
| New | 2,083 (96.1) |
| Retreated** | 84 (3.9) |

**Notes.**

PTB, smear negative pulmonary TB; PTB+, smear positive pulmonary TB; PTBNS, pulmonary TB with unknown sputum; EPTB, extra-pulmonary TB; Underweight*, '5 percentiles; **, relapse, default and failure cases.

## Risk factors of PTB

In univariate analysis, variables which had statistically significant positive association with PTB were being male (OR: 1.329, $p = 0.006$), age group ≤2 years (OR: 2.230, $p < 0.001$) and patients documented as underweight (OR: 1.285, $p = 0.029$) with higher odd ratios comparatively to their counterparts. There was no significant difference observed between rural and urban residents.

After adjusting for risk factors of PTB in univariate logistic regression, variables which were significantly associated with PTB in multivariate analysis were male (OR:

**Table 3  Logistic regression analysis for predictors of PTB ($n = 1,666$).** Bold values indicate significant values.

| Characteristics | PTB | | Univariate analysis | | Univariate analysis | |
|---|---|---|---|---|---|---|
| | Yes *n* (%) | No *n* (%) | COR (95% CI) | *p*-value | AOR (95% CI) | *p*-value |
| **Gender** | | | | | | |
| Male | 771 (79.7) | 197 (20.4) | 1.329 (1.08–1.62) | **0.006** | 1.262 (1.02–1.55) | **0.028** |
| Female | 895 (74.6) | 304 (25.4) | 0.752 (0.61–0.92) | | 0.792 (0.64–0.97) | |
| **Age (years)** | | | | | | |
| ≤2 | 399 (86.6) | 62 (13.4) | 2.230 (1.67–2.97) | **<0.001** | 1.953 (1.36–2.79) | **<0.001** |
| 3–5 | 296 (77.1) | 88 (22.9) | 1.014 (0.78–1.31) | 0.971 | – | – |
| 6–10 | 451 (74.3) | 156 (25.7) | 0.921 (0.66–1.22) | **0.050** | 0.868 (0.64–1.17) | 0.356 |
| 11–14 | 520 (72.8) | 195 (27.3) | 0.712 (0.57–0.87) | **0.001** | 0.820 (0.61–1.09) | 0.181 |
| **Residence** | | | | | | |
| Rural | 715 (77.3) | 210 (22.7) | 1.042 (0.85–1.27) | 0.691 | – | – |
| Urban | 951 (76.6) | 291 (23.4) | 0.960 (0.78–1.17) | | | |
| **Weight (percentiles)** | | | | | | |
| Underweight | 1,271 (78) | 358 (22) | 1.285 (1.02–1.60) | **0.029** | 1.342 (1.06–1.68) | **0.008** |
| Normal | 395 (73.4) | 143 (26.6) | 0.778 (0.62–0.97) | | 0.745 (0.59–0.93) | |

1.262, $p = 0.029$), children aged ≤2 years (OR: 1.953, $p < 0.001$) and those reported as underweight (OR: 1.342, $p = 0.008$) (Table 3).

## Treatment outcomes

Among the 2,167 registered children, 56 (2.6%) had no documented treatment outcomes. Of 97.4% patients with documented treatment outcomes, 92.4% successfully completed treatment. There was no significant difference in the treatment success among male and female patients ($p > 0.05$). Children aged 11 to 14 years had the lowest treatment success (90.2%). The death rate was higher among ≤2 years (2.4%) followed by age group 3 to 5 years with a death rate of 1% (Table 4). The highest rate of default (5.6%) and failure (2.3%) was presented among those aged 11 to 14 years. Female patients had a non-significantly increased rate of default compared to male patients.

Patients from urban areas had the highest proportion of treatment success (94.7%) whereas the rate of failure, default, and death was high among rural and re-treated patients. Correspondingly, among all sputum smear grading, +2 and +3 had lower treatment success rate. Those suffering from EPTB were seen with more successful treatment outcomes relative to those with PTB.

## Factors associated with unsuccessful treatment outcomes

Table 5 presents the risk factors for unsuccessful treatment outcomes in logistic regression analysis. In univariate analysis, age group 11 to 14 years (OR: 1.571, $p < 0.001$), rural residents (OR: 2.101, $p < 0.001$), PTB+ (OR: 3.986, $p < 0.001$), underweight patients (OR: 1.581, $p = 0.033$) and retreated patients (OR: 7.962, $p < 0.001$), were observed as potential predictors of unsuccessful treatment outcomes with higher odd ratios value.

**Table 4  Treatment outcomes of patients as per national guidelines ($n = 2111$).** Bold values indicate significant values.

| Characteristics | Cured $n = 144$ (6.8%) | Completed $n = 1,806$ (85.6%) | Failed $n = 19$ (0.9%) | Default $n = 104$ (4.9%) | Died $n = 22$ (1%) | Transferred out $n = 16$ (0.8%) | Total cases evaluated 2111 (%) | p-value |
|---|---|---|---|---|---|---|---|---|
| **Gender** | | | | | | | | |
| Male | 56 (5.9) | 827 (87.1) | 4 (0.4) | 44 (4.6) | 13 (1.4) | 6 (0.6) | 950 (45) | 0.092 |
| Female | 88 (7.6) | 979 (84.3) | 15 (1.3) | 60 (5.2) | 9 (0.8) | 10 (0.9) | 1,161 (55) | |
| **Age** (years) | | | | | | | | |
| 0–2 | 00 | 421 (92.3) | 00 | 23 (5) | 11 (2.4) | 1 (0.2) | 456 (21.6) | **<0.001** |
| 3–5 | 00 | 345 (93.3) | 2 (0.5) | 16 (4.3) | 4 (1) | 3 (0.8) | 370 (17.5) | |
| 6–10 | 31 (5.3) | 523 (89.2) | 1 (0.2) | 26 (4.4) | 2 (0.3) | 3 (0.5) | 586 (27.8) | |
| 11–14 | 113 (16.2) | 517 (74) | 16 (2.3) | 39 (5.6) | 5 (0.7) | 9 (1.3) | 699 (33.1) | |
| **Residence** | | | | | | | | |
| Rural | 61 (6.8) | 744 (82.6) | 10 (1.1) | 68 (7.5) | 11 (1.2) | 7 (0.8) | 901 (42.7) | **<0.001** |
| Urban | 83 (6.9) | 1,062 (87.8) | 9 (0.7) | 36 (3) | 11 (0.9) | 9 (0.7) | 1,210 (57.3) | |
| **Weight** (percentiles) | | | | | | | | |
| Underweight | 120 (7.5) | 1,343 (84.1) | 18 (1.1) | 85 (5.3) | 19 (1.2) | 11 (0.7) | 1,596 (75.6) | **0.017** |
| Normal weight | 24 (4.7) | 463 (89.9) | 1 (0.2) | 19 (3.7) | 3 (0.6) | 5 (1) | 515 (24.4) | |
| **Type of TB** | | | | | | | | |
| PTB− | 00 | 302 (93.8) | 1 (0.3) | 17 (5.3) | 2 (0.6) | 00 | 322 (15.3) | **<0.001** |
| PTB+ | 144 (66) | 29 (13.3) | 15 (6.9) | 22 (10.1) | 2 (0.9) | 6 (2.8) | 218 (10.3) | |
| PTBNS* | 00 | 1,019 (93.8) | 3 (0.3) | 42 (3.9) | 14 (1.3) | 8 (0.7) | 1,086 (51.4) | |
| EPTB | 00 | 456 (94) | 00 | 23 (4.8) | 4 (0.8) | 2 (0.4) | 485 (23) | |
| **Registration Category** | | | | | | | | |
| New | 130 (6.4) | 1,776 (87) | 14 (0.7) | 96 (4.7) | 15 (0.8) | 11 (0.5) | 2,042 (96.7) | **<0.001** |
| Retreated | 14 (20.3) | 30 (43.5) | 5 (7.2) | 8 (11.6) | 7 (10.1) | 5 (7.2) | 69 (3.3) | |

**Notes.**

PTB, smear negative pulmonary TB; PTB+, smear positive pulmonary TB; PTBNS, pulmonary TB with unknown sputum; EPTB, extra-pulmonary TB.

In multivariate analysis, rural residents (OR: 2.146, $p < 0.001$), PTB+ (OR: 3.409, $p < 0.001$) and re-treated patients (OR: 5.919, $p < 0.001$), were positively associated with unsuccessful treatment outcomes with higher odd ratios.

## DISCUSSION

The present study focused on describing the childhood TB treated under DOTS program in three districts (Hyderabad, Jamshoro, and Matiari) of Sindh province, Pakistan. It reveals the common state of childhood TB support in these districts as the NTP is functioning in the same way throughout the country at all health structures. To the best of our knowledge, this is the very first study presenting epidemiological and clinical data together with clinical outcomes and risk factors for TB in children conducted in the aforementioned districts.

In the present study, childhood TB represented 11.3% of all TB burden at the study site between 2011 and 2015. This was lower than that reported in Tanzania (*Mtabho et al., 2010*). During 2011, the highest proportion of child to adult cases was reported in SGH-QH while for the years 2012 and 2013; LUMHS-CHH had the highest proportion. Additionally, in 2014 and 2015, SBS-THM contributed the highest number of childhood TB cases.

**Table 5 Univariate analysis of risk factors for unsuccessful treatment outcomes ($n = 2{,}111$).** Bold values indicate significant values.

| Characteristics | Treatment success n (%) | Treatment failure n (%) | Univariate analysis | | Multivariate analysis | |
|---|---|---|---|---|---|---|
| | | | COR (95% CI) | p-value | AOR (95% CI) | p-value |
| **Gender** | | | | | | |
| Male | 883 (93) | 67 (7) | 0.861 (0.62–1.19) | 0.369 | – | – |
| Female | 1,067 (92) | 94 (8) | 1.161 (0.83–1.60) | | | – |
| **Age (years)** | | | | | | |
| ≤2 | 421 (92.3) | 35 (7.7) | 1.00 (0.68–1.49) | 0.960 | – | – |
| 3–5 | 345 (93.2) | 25 (6.8) | 0.856 (0.55–1.33) | 0.487 | – | – |
| 6–10 | 554 (94.5) | 32 (5.5) | 0.625 (0.41–0.93) | **0.021** | 0.545 (0.33–0.88) | **0.015** |
| 11–14 | 630 (90.2) | 69 (9.8) | 1.571 (1.13–2.17) | **<0.001** | 0.817 (0.49–1.33) | 0.421 |
| **Residence** | | | | | | |
| Rural | 805 (89.5) | 96 (10.5) | 2.101 (1.51–2.91) | **<0.001** | 2.146 (1.53–3.00) | **<0.001** |
| Urban | 1,145 (94.5) | 65 (5.5) | 0.476 (0.34–0.66) | | 0.481 (0.34–0.67) | |
| **Type of TB** | | | | | | |
| PTB⁻ | 302 (93.8) | 20 (6.2) | 0.774 (0.47–1.25) | 0.302 | – | – |
| PTB⁺ | 173 (79.4) | 45 (20.6) | 3.986 (2.73–5.81) | **<0.001** | 3.409 (2.11–5.48) | **<0.001** |
| PTBNS* | 1,019 (93.8) | 67 (6.2) | 0.653 (0.47–0.90) | **0.010** | 1.072 (0.68–1.68) | 0.762 |
| EPTB | 456 (94) | 29 (6) | 0.727 (0.47–1.09) | 0.126 | – | – |
| **Weight (percentiles)** | | | | | | |
| Underweight | 1,463 (91.7) | 133 (8.3) | 1.581 (1.03–2.40) | **0.033** | 1.441 (0.91–2.23) | 0.102 |
| Normal weight | 487 (94.6) | 28 (5.4) | 0.632 (0.41–0.96) | | 0.694 (0.44–1.07) | |
| **Registration category** | | | | | | |
| New | 1,906 (93.7) | 136 (6.3) | 0.125 (0.07–0.21) | **<0.001** | 0.17 (0.09–0.30) | **<0.001** |
| Retreated | 44 (60.7) | 25 (39.3) | 7.962 (4.73–13.40) | | 5.919 (3.36–10.40) | |

**Notes.**

PTB, smear negative pulmonary TB; PTB+, smear positive pulmonary TB; PTBNS, pulmonary TB with unknown sputum; EPTB, extra-pulmonary TB.

Patients aged ≤2 years constituted 21.3% of total cases despite the fact that the risk of TB is greatest in this age group (*Marais et al., 2004*). The highest cases of TB in the current study were observed among the children aged 11 to 14 years (33%) which is in agreement with the previous studies conducted in Pakistan (*Safdar et al., 2010*) and Ethiopia (*Hailu, Abegaz & Belay, 2014*; *Tilahun & Gebre-Selassie, 2016*). The most reasonable explanation can be difficulties in the diagnosis of younger children, principally in collecting bacteriologic specimens (*Marais et al., 2006*) or gastric aspirates (*Planting et al., 2014*; *Zar et al., 2005*) which can result in under reporting of TB cases in this age group.. The notification rate of childhood TB was higher in urban areas (57.3%) than in rural areas, which conforms to a study from Northwest Ethiopia (*Tessema et al., 2009*). The lower incidence rate in rural areas might possibly be due to under-diagnosis or limited access to the treatment centres. In addition, this might be due to illiteracy and lack of awareness of TB among caregivers/parents.

PTB was diagnosed in 76.9% of children, of which 221 (13.3%) were bacteriologically confirmed PTB+ cases and 41 of these were registered as re-treatment cases. However, greater numbers of PTB+ cases among children are reported from India and Africa (*Aketi et*

*al., 2016*; *Satyanarayana et al., 2010*). Greater numbers of cases were diagnosed on clinical background as they were unable to expectorate and techniques like induced sputum and gastric lavage to acquire sputum among young children for smear microscopy are not frequently used at the study site. The proportion of EPTB in the current study was 23.1%. The proportion was parallel to the studies conducted in other areas of Pakistan (*Batra et al., 2012*; *Safdar et al., 2010*) and Taiwan (*Tsai et al., 2013*), but lower than that observed in Ethiopia (*Hailu, Abegaz & Belay, 2014*) and southern Taiwan (*Cho et al., 2014*). TB was slightly higher in female children (55.3%) compared to males (44.7%). This finding is in line with previous studies (*Sharma et al., 2008*; *Tilahun & Gebre-Selassie, 2016*). Unfortunately, in some areas Pakistan, girls have an inferior status and have limited rights, prospects and benefits of childhood than boys. At a very young age, experiences of inequality are initiated for women and it is very difficult for them to overwhelm this unfairness. In general, the girl gets less food, poorer access to education and less health care than boys, particularly in families with a poor socio-economic background. A remarkable proportion of the present study participants were underweight (75.2%). According to WHO, malnutrition is a pre-defined risk factor of TB in children, i.e., children with TB are generally found to be malnourished. Malnutrition is prevalent among all ages in Pakistan. Unfortunately, one-third of children under 5 years old in Pakistan are reported underweight, 44% as stunted and 15% wasted (*Das, Achakzai & Bhutta, 2016*). A high prevalence of malnourishment in patients of the current cohort was a result of their socio-economic background.

Successful outcome is a marker of the quality of TB case management (*WHO, 2014*). The overall TSR of 92.4% in the present study met the WHO target of 85%. The TSR is higher than the TSR reported in Malawi (*Harries et al., 2002*) and Botswana (*Oeltmann et al., 2008*) but lower than that reported in Delhi, India (*Satyanarayana et al., 2010*). Treatment outcomes were recorded for 2,042 new cases and 69 re-treatment cases. There were 56 children for whom the treatment record was not available. The death rate in this study (1.0%) is comparable to that in Punjab, Pakistan (*Safdar et al., 2010*) and India (*Satyanarayana et al., 2010*). In 2014, 1 million new cases of childhood TB were estimated by WHO with 136,000 deaths at the proportion of 13.6% at the global level (*Jenkins, 2016*). However, the death rate in the present study was much lower than death rates (3.3–17%) previously reported elsewhere (*Dangisso, Datiko & Lindtjørn, 2015*; *Hailu, Abegaz & Belay, 2014*; *Harries et al., 2002*; *Mtabho et al., 2010*; *Oeltmann et al., 2008*). The default rate (4.9%) encompassed the foremost part of the unsuccessful outcome and was higher than that (0.6–3.8%) reported from other regions (*Hailu, Abegaz & Belay, 2014*; *Satyanarayana et al., 2010*; *Tilahun & Gebre-Selassie, 2016*) but lower (7%) than that informed by (*Aketi et al., 2016*).

In multivariate analysis, rural areas, sputum smear positivity, underweight, and re-treatment were significantly associated with poor treatment outcomes. These findings are in line with previous studies (*Bloss et al., 2012*; *Jaganath & Mupere, 2012*; *Sharma et al., 2008*). The most important reason for all these risk factors could be non-adherence due to lack of education among the caregivers. Sindh is a multicultural province of Pakistan and is portrayed by an extensive gap between rich and poor individuals with unequal access to education and health services. Insufficient knowledge about TB and its treatment, longer

distances to treatment centre and associated costs affect patients' access to TB care and clinical outcomes result in treatment delays and poor treatment outcomes. Moreover, children reported underweight were observed as at a high risk of TB in the study area that signposted the poverty ratio and socioeconomic condition of the family that is an imperative issue for which instant steps need to be taken by the authorities.

### Limitations of study

The study has some limitations. Firstly, since the study was retrospective it was not possible to collect socioeconomic data and adverse effects observed during the treatment. Factors like family income and parents' education, that could affect the treatment outcomes were not recorded and thus not examined. Secondly, features such as household size, household contacts and nutritional status, which might serve as risk factors for childhood TB, were also not documented.

## CONCLUSIONS

In the present study, childhood TB represented 11.3% of all TB burden at the study site. The treatment outcomes of childhood TB treated under the DOTS program in the study area were satisfactory. The fairly high success rate could be due to enhanced case management with the accessibility of free TB treatment and the promising performance of the DOTS strategy at the health centres in study. Children aged $\leq 2$ years and male, living in rural areas, sputum smear positive, underweight and those who are being re-treated should be given special attention as they have a significant association with TB and poor treatment outcomes.

**Abbreviations**

| | |
|---|---|
| **AFB** | Acid-Fast Bacilli |
| **ATT** | anti-tuberculosis treatment |
| **AORs** | adjusted odds ratios |
| **COR** | crude odds ratios |
| **CXR** | chest X-ray |
| **CIs** | confidence intervals |
| **DOTS** | directly observed therapy strategy |
| **EPTB** | extra-pulmonary TB |
| **FNAC** | Fine Needle Aspiration Cytology |
| **Xpert MTB/RIF** | Gene Xpert MTB/RIF |
| **ICDK** | Institute of Chest Diseases Kotri, Sindh |
| **LUMHS-CHH** | Liaquat University of Medical and Health Sciences Civil Hospital, Hyderabad |
| **MTB** | *Mycobacterium tuberculosis* |
| **NTP** | National Tuberculosis Control Program |
| **PTB** | pulmonary TB |
| **PTBNS** | pulmonary TB with unknown sputum |
| **PTB−** | smear negative PTB |
| **PTB+** | smear positive PTB |

| | |
|---|---|
| **SBGH-LH** | Shah Bhitai Government Hospital, Latifabad, Hyderabad |
| **SBS-THM** | Syed Baqadar Shah Taluka Hospital, Matiari |
| **SGH-QH** | Sindh Government Hospital Qasimabad, Hyderabad |
| **TB** | tuberculosis |
| **TST** | Tuberculin skin test |
| **WHO** | World Health Organization |

### Funding
The authors received no funding for this work.

### Competing Interests
The authors declare there are no competing interests.

### Author Contributions
- Madeeha Laghari conceived and designed the experiments, performed the experiments, prepared figures and/or tables.
- Syed Azhar Syed Sulaiman conceived and designed the experiments, contributed reagents/materials/analysis tools, approved the final draft.
- Amer Hayat Khan analyzed the data, authored or reviewed drafts of the paper.
- Naheed Memon authored or reviewed drafts of the paper, approved the final draft.

### Human Ethics
The following information was supplied relating to ethical approvals (i.e., approving body and any reference numbers):

This study was conducted with the permission of the relevant Institutional Research and Ethics Boards (IREBs) of Shah Bhitae Hospital Latifabad, Hyderabad Liaquat University Hospital Hyderabad / Jamshoro, Sindh Government Hospital Qasimabad, Hyderabad, Sayed Baqadar Shah Civil Hospital Matiari and Institute of chest diseases Kotri Sindh, Pakistan (Vide Letter No: SBGH/L.ABAD HYD-1575; Dated: 13-04-17, LUH/Estt/-23176/14; Dated: 06-08-2016, MS-SGHQ/HYD/2187: Dated:13-04-17, CS/CH/MAT:1761; Dated:18-05-16 and ICDK/771; Dated: 12-04-17, respectively).

### Data Availability
The raw data are provided in Data S1.

### Supplemental Information
Supplemental information for this article can be found online at http://dx.doi.org/10.7717/peerj.5253#supplemental-information.

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
