# Peer review of "Epidemiology of tuberculosis and treatment outcomes among children in Pakistan: a 5 year retrospective study"

_PeerJ, doi:10.7717/peerj.5253_

## Round 0.1 · original submission · Major Revisions

This manuscript addresses the important concerns with pediatric TB. I do believe this study is important and has a significant contribution to understanding TB in children. However, the manuscript can be improved further and can be crisp. I agree with the reviewers and suggest the authors to pay careful attention to the concerns raised by the reviewers.

Please spell check (esp for WHO) the manuscript and also add references to statements made both in introduction and discussion. some key statements are missing references. please check the references for uniformity as well.

I will look forward to receiving your revised manuscript. Thank you.

Reviewer 1 ·

Basic reporting

1. There is noted ambiguity in lines 99 to 114, 175 to 176, and lines 232 to 239.
2. Language used could be improved in lines 263 to 270 (unsure if personal opinion, which is inappropriate, or if is from references listed in sentences preceding. Language improvement also needed Lines 255 to 260.
3. Introduction adequate and can appreciate the background intended by author. However, the reference citing need to be improved. These are apparent in line 50 where TB is actually the nineth leading cause of death globally and NOT by an infectious disease. The points of attempting to evaluate the burden of paediatric TB disease in this area within a high burden region is noted but not consistent following intro.
4. Structure does more or less conform to PeerJ standards but references need to be looked at further i.e.between journal title and volume needs to be amended.
5. Figures are relevant except the the weight band categories figure is not elaborated upon and confusing at best. I suggest you review the WHO new FDC formulations and weight band categories and try to compare those against standard of available care with regard to treatment regimens available in Pakistan. Good quality explanatory figures explaining trends over fixed period. Figures should have descriptive comments written below, whilst tables should have the title of the table typed above the table.
6. Raw data seems fine.

Experimental design

1. The primary research and evaluation of the district in question is within the scope of the journal and is original, highlighting the need to investigate areas of possible concern within a high burden area.
2. The research question is well defined and pertinent toward the global context of childhood TB.
3. There seems to be no IRB ethical approval; no mention of child assent or informed parental/guardian consent; unsure if data available were de-identified to ensure subject confidentiality. Only site approvals from medical superintendents. Investigation tools used seem appropriate but there seems be poor application of paediatric research and clnical definitions.
4. Noted methods of retrospective analysis of data from TB register. Would you call this a retrospective cohort study?
The inclusion criteria is poorly stated and ambigious as mentioned above in “basic reporting” and suggest you review current up to date or recommended Paediatric TB definitions as well as others eg Graham et al (J infect Dis, 2012, V205, pp 199) or Seddon et al (J of Paed Infect Dis, 2013, V2, pp 100).
Childhood symptomatology are different from adult TB and you aren’t clear about this. The inclusion criteria makes no mention of microscopy for smear status, Xpert availability as a WHO recommended screen for children and phenotypic culture as a gold standard still adhered to for childhood TB diagnosis. Need to be clear about why clinical diagnoses made more.
5. Personally I feel WHO malnutrition anthropometric defintions might be more useful for a global reader audience i.e. with Z-scores and defining “moderate acute malnutrition” and “severe acute malnutrition” but I do note that you only had TB register data available to evaluate. This is not a must have as is retrospective data analysis.

Validity of the findings

1. The incidence of childhood TB globally is mentioned at roughly 5% in the introduction (line 72). No mention of TB high burden region sttaifications by WHO and Pakistan, as part of the Eastern Meditarranean high burden region. Mention of incidence and no mention of prevalence of TB childhood disease in pakistan over a known time period from existing literature and to use that as a backbone for relating current burden of disease, morbidity and mortality in context. Rather your referencing has not kept in mind WHO regions and of a recent Lancet meta-analysis by Jenkins et al reviewing global childhood TB morbidity and mortality to corelate against your findings in discussion.
2. Risk factors for Tb mentioned are in keeping with norms in clinical experience and agreed with.
3. Mention again: for rather looking at recent meta-analysis of TB regions wrt childhood TB.
4. Line 239 paraphrased states a gold standard “needed to be developed”. This is inaccurate as there is a gold standard with certain limitations which could be elaborated upon – ie phenotypic culture usually MGIT culture.
5. Line 264 to 270 seems more a personal opinion and needs to be reworded and shown to be of evidence-base.
6. It is well known that only half of children with TB disease are diagnosed and put on treatment (WHO Global Report, 2016). Those that are inititated on treatment appropriately do very well usually.
7. Data presented seems statistically reasonable to assess. But if the global burden of childhood TB is roughly 6.9% and we are unsure if over- or under diagnosing, why are your findings across the 3 sites evaluated not improving drastically and high proportionately?
8. I don’t think the conclusion is well stated and doesnt bring your research question to fruition in terms of local and global context.

Additional comments

1. It is a worthy study to have undertaken and glad that data is avaliable to review how these sites contribute to the regional burden of childhood TB in the Eastern Meditarreanean region as well as in the global scene.
2. Need to mention if IRB ethical approval has been granted.
3. Reference citing and understanding of current paediatric TB clinical and epidemiological needs work.
4. Some citations are old evidence and newer, updated revised evidence is available.
5. Conclusion doesn’t quite coalesce well.

·

Basic reporting

The manuscript “Epidemiology of tuberculosis and treatment outcomes among children in Pakistan: A 5 year retrospective study (#24857)” by Laghari et al presents interesting data and conclusions on the epidemiology of TB in children in Pakistan.

It is the opinion of the reviewer that the manuscript can be accepted for publication with minor corrections.

Experimental design

no comment

Validity of the findings

no comment

Additional comments

- Did the authors observed if there was any difference in the treatment success rate between patients infected with MDR strains of TB and patients infected with strains sensitive to drugs? This could be an interesting data.

- Line 82-83: Since the conclusions of the manuscript indicates that there is a significant difference in the success rate of the treatment between children living in rural and urban areas, can the authors include a short comment indicating the major differences in living style and condition between rural and urban areas?
- Line 123: Please clarify the treatment regimens reported in Table 1. It is not clear
- The authors should indicate s the antibiotic treatment regime advised in Pakistan.

- Table 1: Please indicate the meaning of the abbreviations “R”, “H”, “Z” and “E”
- Figure 2: please indicate what the abbreviations stand for.

---

## Round 0.2 · Minor Revisions

Thank you for your resubmission. the manuscript is much improved and reads better. Please address the minor revisions requested by the reviewer 1.

Personally, I have few comments that I would like to add

1. background is very lengthy and redundant. Please consider making it as precise as possible. So, is the discussion. there are many places you can combine sentences and remove over explanations.
2. Line 112. double 'was' in the sentence.
3. please define " treatment success and treatment failure" in the text. Since there is no universal definition for these terminologies, it is better to make it clear with reference to your study.
4.please add country name following the city name. for example in lines 323, 326, you mention Delhi but no country name, Punjab, again no country name. since these places are in multiple countries it is easier for the readers to understand rather than going back to the references.
5. spelling for 'Organization'.. please check and revise.

Reviewer 1 ·

Basic reporting

1. Must commend the authors on a much improved manuscript.
2. Introduction good. Minor comment: Language line 62 “This is for the reason” can be improved for an international reader audience.
3. Minor Language comment in Line 263 “Like” could be changed.
4. Figures and tables much improved. Unsure if PeerJ would like Figure captions to be below figure and have Table headers above table as shown.

Experimental design

1. Recommendations addressed and am happy with design.
2. Re: “suggest you review current up to date or recommended Paediatric TB definitions as well as others eg Graham et al (J infect Dis, 2012, V205, pp 199) or Seddon et al (J of Paed Infect Dis, 2013, V2, pp 100).”
i. Most paediatric Tb publications used recommended definitiions as part of academic consensus for terms labelled “Definite Tb, Probable TB, Possible TB” or “definite, probable or not TB.”
ii. I’m happy with the authors rebuttal but a comment to authors for reader audience interested in their findings.

3. Wrt my previous comment re: “Personally I feel WHO malnutrition anthropometric defintions might be more useful for a global reader audience i.e. with Z-scores,” I must apologise to the authors.
(i) WHO anthrometric gradings with centile and z scores (as used in South African paediatric clinical settings) theoretically are not as appropriate to be used in the developing world setting as the WHO growth charts are more targeted to a healthy index population.
(ii) The CDC definitions are more appropriate.

Validity of the findings

1. Re: "Rather your referencing has not kept in mind WHO regions and of a recent Lancet meta-analysis by Jenkins et al reviewing global childhood TB morbidity and mortality to corelate against your findings in discussion." – I still would have liked the authors to have read an article and refer them to:
i. Stark et al (Lancet Infect Dis, 2016, V 17, pp 239) for commentary on Helen Jenkins article on Global childhood TB mortality.
ii. This gives context to their article and think the authors need to realize that the article targets a paediatric academic community also, esp the paediatric TB academic community.
2. Mention again: for rather looking at recent meta-analysis of TB regions wrt childhood TB. – see comment 1.
3. With reference to known paediatric TB epidemiological trends I am unhappy with the explanation in lines 245 to 250 (not explained well and seems not appreciative of disease spectrum reasoning behind it).
i. Trends show that with adolescence there an increased trend of TB disease in this age group presenting with more adult type of disease spectrum. Younger children have paucibacillary pathogen burden while older children much higher burden
ii. There is also hypotheses that adult sex hormones via the HPG axis might account for these.
iii. I think these lines should be looked at carefully and look at epidemiological age curves demonstrated from various publications incl WHO. Perhaps can look at articles by Peter Donald too.
4. Line 278 to 285 seems more a personal opinion still. Is there no qualitative research evidence to support this to make your point more poignant and contextual? Noted comment on being a hypothesis so perhaps state it clearer that its specultive in terms of personal opinion; it is quite a long explanation describing the hypothesis.
5. Data presented seems statistically reasonable to assess. I am happy with the authors rebuttal.
6. Conclusion sounds good.

Additional comments

1. I am very happy with the revisions made and commend the authors for the hard work put in.
2. My comments arise from being a paediatrician and a childhood TB researcher. Comments were only geared to help you target an interested audience.

·

Basic reporting

The authors have replied to all the comments from the reviewers and the article can now be accepted for publication

Experimental design

no additional comments

Validity of the findings

no additional comments

Additional comments

As I mention in my previous review, the article present interesting results about epidemiology of TB in children in Pakistan. The authors replied to the comments of the reviewers and the article now can be accepted for publication.

---

## Round 0.3 · accepted · Accept

Dear Dr. Laghari

Thank you for the resubmission. I recommend the manuscript for accept contingent upon addressing the minor revisions requested by the reviewer. Congratulations.

Best wishes